# Multilingual Contextualization of Large Language Models for Document-Level Machine Translation

**Miguel Moura Ramos**[1,2]  **Patrick Fernandes**[1,2,3]  **Sweta Agrawal**[2]
**André F. T. Martins**[1,2,4]
[1]Instituto Superior Técnico, Universidade de Lisboa (ELLIS Unit Lisbon)
[2]Instituto de Telecomunicações  [3]Carnegie Mellon University  [4]Unbabel

## Abstract

Large language models (LLMs) have demonstrated strong performance in sentence-level machine translation, but scaling to document-level translation remains challenging, particularly in modeling long-range dependencies and discourse phenomena across sentences and paragraphs. In this work, we propose a method to improve LLM-based long-document translation through targeted fine-tuning on high-quality document-level data, which we curate and introduce as DOCBLOCKS. Our approach supports multiple translation paradigms, including direct document-to-document and chunk-level translation, by integrating instructions both with and without surrounding context. This enables models to better capture cross-sentence dependencies while maintaining strong sentence-level translation performance. Experimental results show that incorporating multiple translation paradigms improves document-level translation quality and inference speed compared to prompting and agent-based methods.

## 1 Introduction

Large language models (LLMs) have demonstrated strong performance across a wide range of natural language processing tasks (Ouyang et al., 2022; Sanh et al., 2021; Wei et al., 2021; Touvron et al., 2023; Jiang et al., 2023), including machine translation (MT) (Zhang et al., 2023; He et al., 2024; Hendy et al., 2023). Recent work (Alves et al., 2024; Xu et al., 2023) has convincingly shown that LLM-based systems consistently outperform specialized encoder-decoder MT systems for many language pairs (Kocmi et al., 2024; Deutsch et al., 2025). However, the focus of these studies has been primarily on sentence- and paragraph-level translation, overlooking the complexities of translating entire documents where maintaining coherence, consistency, and discourse structure is fundamental. For document-level translation, various techniques have emerged, such as context-aware prompting (Wang et al., 2023a; Wu et al., 2024) and agent-based translation strategies (Wu et al., 2024b; Wang et al., 2024). On the other hand, supervised fine-tuning (SFT) has proven highly effective for improving sentence-level MT (Alves et al., 2024; Xu et al., 2023), but its adaptation to document-level translation, and its comparison with other techniques, remain open questions.

In this work, we take a fresh perspective on the problem and explore whether we can employ SFT on strong sentence-level LLMs (Alves et al., 2024; Xu et al., 2023; Martins et al., 2024) to adapt them for multilingual contextualization. Specifically, we tackle the underexplored challenge of training LLMs to leverage surrounding context for document-level translation. Towards this end, we introduce DOCBLOCKS, an MT contextualization dataset carefully curated and filtered to incorporate high-quality document-level data. Unlike prior datasets (NLLB Team et al., 2022; Kocmi et al., 2023; 2024) that focus primarily on isolated sentence pairs or limited context windows, DOCBLOCKS consists of full documents and contextual segments sourced from trusted parallel corpora across diverse domains, including news, TED talk transcripts (Cettolo et al., 2017), literary texts, and parliamentary proceedings. We then introduce a simple yet effective contextualization method that fine-tunes existing LLM-based MT systems on DOCBLOCKS by integrating the data in multiple formats. These include full documents, contextualized document chunks with preceding sections (Wang

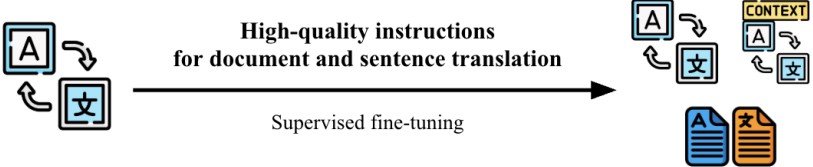

Figure 1: Illustration of our approach for adapting an LLM to document translation.

et al., 2023a) and standalone sentence-level examples. This multi-granular training strategy enables the model to capture both document structure and inter-sentence relationships, improving document-level translation performance while maintaining its robust sentence-level capabilities (Figure 1). Finally, to evaluate the dependence of our approach on the base MT model, we experiment with three different LLM-based MT models: TOWER (Alves et al., 2024), EUROLLM (Martins et al., 2024), and QWEN2.5 (Bai et al., 2023). This allows us to assess the generalizability of our method across various model architectures.

Our key contributions are summarized as follows:

- 🧩 **DOCBLOCKS dataset**[1]: We release a document-level parallel dataset for tailoring LLMs to document translation, designed to support long-range dependency modeling and discourse coherence.

- ⚙️ **Fine-tuning for Multilingual Contextualization**: We propose an efficient fine-tuning approach that improves the document-level translation capabilities of LLMs by training on diverse, high-quality instructions across various translation paradigms—improving contextual understanding while preserving sentence-level performance.

- 📊 **Comprehensive Evaluation**: We conduct a systematic comparison of fine-tuning, prompting, and agent-based methods for document-level MT. Additionally, we perform ablation studies to evaluate the individual components of our approach, demonstrating its effectiveness in tailoring LLMs for document translation.

Our experiments across multiple datasets and language pairs demonstrate that our fine-tuning approach, combined with lightweight prompt-based context modeling, significantly improves document-level translation performance across various paradigms, outperforming vanilla LLM-based MT models, as well as both prompting and agent-based techniques.

## 2 Multilingual Contextualization of LLMs for Document-level MT

In this section, we present our contributions, starting with DOCBLOCKS, a high-quality parallel dataset curated to address data scarcity in document-level MT (§2.1). Next, we outline the fine-tuning process of LLMs using this dataset to enhance the performance of document-level machine translation (§2.2). Finally, we investigate various inference methods and assess their impact on document translation quality (§2.3).

### 2.1 🧩 DOCBLOCKS: A High-Quality Document-level MT Parallel Corpora

**Corpora Collection.** The DOCBLOCKS corpus is constructed from a collection of publicly available document-level datasets, selected to represent a broad range of document types and content domains. Figure 2 summarizes the source datasets and their key characteristics. The primary objective of developing DOCBLOCKS was to compile a versatile corpus that enables effective training of LLMs for document-level MT tasks. The *News Commentary* corpus contains political and economic news texts and was included as a representative of

---

[1]Available on Hugging Face.

| Domain | Corpus | Pre-filtering | | | | Post-filtering | | | |
|---|---|---|---|---|---|---|---|---|---|
| | | \|D\| | \|S\| | \|W\| | \|W\|/\|D\| | \|D\| | \|S\| | \|W\| | \|W\|/\|D\| |
| *news* | News Commentary v18.1 | 125.0K | 4.9M | 119.0M | 1.0K | 110.0K | 4.4M | 96.4M | 0.9K |
| *TED Talks* | IWSLT2016/2017 | 33.5K | 3.3M | 82.4M | 2.5K | 29.7K | 2.6M | 64.0M | 2.1K |
| *Parliamentary* | Europarl v10 | 44.8K | 22.3M | 542.0M | 12.1K | 21.3K | 10.5M | 264.8M | 12.4K |
| *Novels* | GuoFeng | 22.6K | 1.9M | 51.4M | 2.3K | 18.2K | 1.7M | 28.5M | 1.6K |
| | BWB | 196.0K | 10.3M | 450.0M | 2.3K | 62.3K | 4.2M | 110.3M | 1.8K |

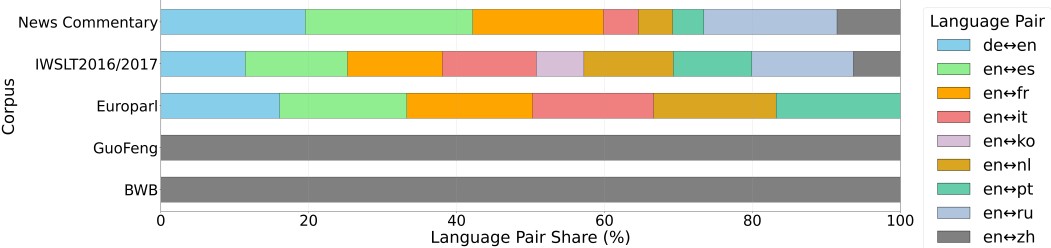

Figure 2: Statistics of datasets used to create DOCBLOCKS, including domain, corpus name, number of documents $|D|$, sentences $|S|$, words $|W|$, and average document length $|W|/|D|$ as a proxy for discourse complexity (Wang et al., 2023a). We also illustrate the distribution of language pairs across the filtered datasets, highlighting their multilingual composition.

well-structured written content from the journalistic domain. The **IWSLT** (Cettolo et al., 2017) corpus, based on TED Talk transcripts, provides examples of conversational and spoken language across diverse topics. The **Europarl** (Koehn, 2005) corpus consists of European Parliament proceedings and represents formal writing with longer sentence structures and extended document length, typical of institutional discourse. The **BWB** (Jiang et al., 2022) and **GuoFeng Webnovel** (Wang et al., 2023b) corpora were included for their expert translations of Chinese novels and web fiction, which feature complex discourse structures and genre diversity. We also considered the **United Nations Parallel Corpus** (Ziemski et al., 2016) and **OpenSubtitles** (Lison & Tiedemann, 2016) for their multilingual scope and size, but document segmentation, alignment issues, and increased computational costs led us to prioritize better-curated datasets.

**Data Curation: Preprocessing, Cleaning, and Augmentation.** All datasets consist of document-level parallel corpora. We do not enforce strict sentence-to-sentence alignment. Instead, alignment is maintained at the document level, allowing for variation in sentence count and structure – reflecting realistic conditions in document translation. This approach enables the model to rely on broader semantic context rather than rigid sentence mappings. For datasets containing paragraph-level content with document boundary metadata, we grouped paragraphs into full documents and concatenated them up to a maximum length of 32,768 tokens. To ensure high-quality data, we applied a rigorous cleaning pipeline to DOCBLOCKS. Low-quality translations were identified using automatic filters, including Bicleaner (Ramírez-Sánchez et al., 2020) (threshold: 0.5) and CometKiwi-23 (Rei et al., 2023) (threshold: 0.65). We removed entire documents if over 20% of their sentences fell below these thresholds. We further applied language identification with langid (Lui & Baldwin, 2012) to exclude misaligned language pairs. Additional filters removed documents with fewer than 50 words, more than 10% identical characters/words/numbers, or a source–target length ratio above 1.3. We also deduplicated all entries.

These steps yielded a clean, reliable dataset optimized for document-level translation. To further enhance learning, we incorporated two complementary training techniques:

1. *Multi-Resolutional Document-to-Document Training (MRD2D)* (Sun et al., 2022): We split each document into $k$ parts where $k \in \{1, 2, 4\}$ to create sequences of varying lengths, improving computational efficiency.

2. *Context-Aware Prompt Tuning (CAPT)* (Wang et al., 2023a): Building on the same preprocessed chunks, we incorporate a context window of up to the previous 3 segments in each training prompt. This allows the model to better capture document-level dependencies during training.

We applied MRD2D and CAPT selectively to datasets with longer average document lengths and moderate size, specifically, IWSLT, Europarl, and GuoFeng, to manage data growth and training costs. Experiments show that this curation and training strategy significantly improves translation performance (see Table 6).

## 2.2 Document-Level Fine-Tuning with DOCBLOCKS

**Model Training.** Following prior work (Zhang et al., 2018; Wu et al., 2024), we adopt a two-step learning strategy, where LLMs are first tuned for sentence-level MT and then adapted for document-level MT tasks. Given the availability of sentence-level instruction-tuned LLMs (Alves et al., 2024; Martins et al., 2024), we focus only on the second step, which further fine-tunes these models on document-level translation data. To ensure high-quality translations, we calculate the loss exclusively on target tokens, excluding prompt tokens (source and instruction tokens), thereby preventing penalties for prompt format adherence.

**DOCBLOCKS instruction format.** We support three instruction formats using the `chatml` (OpenAI, 2023) template. Following Wu et al. (2024), we use a contextual block of up to $N = 3$ previous chunks and format the texts as shown in Table 1. In addition to the chunk-level format, we also include instruction examples for document-to-document and sentence-to-sentence translation tasks. These tasks follow a simpler instruction structure, without the contextual block.

| User | `<|im_start|>user` |
|---|---|
| | Context: |
| | {source_lang}: {source$_1$}  {target_lang}: {target$_1$} |
| | {source_lang}: {source$_2$}  {target_lang}: {target$_2$} |
| | {source_lang}: {source$_3$}  {target_lang}: {target$_3$} |
| | Translate the following source text from {source_lang} into {target_lang}. |
| | {source_lang}: {source}. |
| | {target_lang}: `<|im_end|>` |
| | `<|im_start|>assistant` |
| Model | {target}. `<|im_end|>` |

Table 1: Illustration of a DOCBLOCKS instance, with the contextual block in gray.

## 2.3 Inference

We investigate the impact of different inference methods on document translation performance, as our models are designed to support a variety of translation strategies. We explore two methods: **Document-to-Document (Doc2Doc)** and **Chunking**.

The *Doc2Doc* method translates the entire document in a single pass, leveraging the ability of the LLM to capture long-range context. When the document fits within the model context window, it typically yields the most coherent and consistent translations (Wang et al., 2023a).

The *chunking* method translates a document chunk by chunk, with each chunk comprising a fixed number of sentences, paragraphs, or tokens. It can be applied on its own or enhanced in two complementary ways that address different aspects of the translation process. First, *contextual chunking* improves coherence by conditioning the translation of each chunk on previously translated source-target chunk pairs. Second, *quality-aware chunking* focuses on selecting the best translation using Minimum Bayes Risk (MBR) decoding (Kumar & Byrne, 2004; Eikema & Aziz, 2022), guided by quality estimation metrics such as COMET (Rei et al., 2022) and CONTEXTCOMET (Vernikos et al., 2022). These metrics may incorporate contextual information and help choose the final output by maximizing expected utility:

$\hat{y}_{mbr} = \arg\max_{y \in \bar{\mathcal{Y}}} \mathbb{E}_Y [u(Y, y)] \approx \arg\max_{y \in \bar{\mathcal{Y}}} \frac{1}{|\bar{\mathcal{Y}}|} \sum_{y_t \in \bar{\mathcal{Y}}} \mathcal{M}([c_t, y_t], [c, y])$, where $\bar{\mathcal{Y}}$ is a list of candidates sampled by the model and $\mathcal{M}$ is the reference-based quality metric, which can optionally use context $c$.

## 3 Experiments

### 3.1 Experimental Setup

**Datasets.**  We evaluate translation performance using several benchmarks. For document-level MT, we use IWSLT2017 (Cettolo et al., 2017) and GuoFeng (Wang et al., 2023b) test sets. IWSLT2017 contains parallel TED talk documents across six language pairs: en↔{de, fr, it, ko, nl, zh}. GuoFeng, a discourse-rich web novel corpus, is used for zh→en experiments on its combined simple and difficult test sets. For sentence-level MT, we use FLORES-200 (en↔{de, fr, it, ko, nl, zh, ru, pt, es}) (NLLB Team et al., 2022), WMT23 (en↔{de, ru, zh}) (Kocmi et al., 2023), and TICO-19 (en→{es, fr, pt, ru, zh}) (Anastasopoulos et al., 2020) to assess how document-level training affects sentence-level MT quality.

**Baselines.**  In this work, we employ the closed-source GPT-4O (OpenAI et al., 2023) and the open-source models QWEN2.5-72B-INSTRUCT (Bai et al., 2023) and LLAMA-3.3-70B-INSTRUCT (Grattafiori et al., 2024) as our baselines. These three state-of-the-art LLMs serve as the primary comparisons for evaluating our approach to adapting sentence-level LLMs to document-level MT. We use greedy decoding for all inference methods, except for quality-aware chunking, where we generate 32 candidates using $p$-nucleus sampling (Holtzman et al., 2019) with $p = 0.6$.

**Base sentence-level LLMs.**  To assess the effectiveness of our method across different base models, we experiment with three instruction-tuned LLMs: TOWERINSTRUCT-MISTRAL-7B (Alves et al., 2024), EUROLLM-9B-INSTRUCT (Martins et al., 2024), and QWEN2.5-7B-INSTRUCT (Bai et al., 2023). While TOWERINSTRUCT-MISTRAL-7B offers a 32768 context window, it struggles with document-level MT due to its pretraining and instruction tuning, which are focused on short sequences. EUROLLM-9B-INSTRUCT has a 4096 context window, limiting its performance in document-level MT. In contrast, QWEN2.5-7B-INSTRUCT performs better with its 32768 context window (see Tables 2 and 3).

**Hyperparameters.**  We fine-tune all sentence-level LLMs on the instructional DOCBLOCKS using the standard cross-entropy loss and enable `bfloat16` mixed precision and packing (Raffel et al., 2020). We detail all hyperparameters in Appendix A. For TOWERINSTRUCT-MISTRAL-7B and QWEN2.5-7B-INSTRUCT, we kept the original RoPE theta (Su et al., 2021) since both were pretrained with a 32768 context window. For EuroLLM, initial tests showed minimal gains from extending RoPE, so we retained the original design.

**Evaluation.**  We evaluate different approaches using both sentence-level and document-level metrics. Sentence-level evaluation includes BLEU (Papineni et al., 2002) and COMET (`wmt22-comet-da`) (Rei et al., 2022). Sentence alignments are first computed using `bleualign`, after which these metrics are applied. However, these metrics have known limitations in capturing document-level phenomena such as coherence, cohesion, and referential consistency (Castilho et al., 2020; Fernandes et al., 2021; Läubli et al., 2018; Toral, 2020; Freitag et al., 2021). To better evaluate document-level translation quality, we report document-level BLEU (d-BLEU) (Papineni et al., 2002; Liu et al., 2020) with corresponding brevity penalty (BP), and document-level COMET (d-COMET) (Rei et al., 2022), computed using the SLIDE approach (Raunak et al., 2024). SLIDE divides each document into overlapping 512-token chunks (the context limit for COMET), computes chunk-level scores independently, and averages them to obtain a document-level score. We also report the Lexical Translation Consistency Ratio (LTCR) (Wang et al., 2023a; Wang et al., 2024; Xiao et al., 2011; Lyu et al., 2021), which captures the consistency of repeated terminology across a document. Although recent work has proposed alternative document-level metrics (Vernikos et al., 2022; Jiang et al., 2022; Gong et al., 2015; Wong & Kit, 2012), there remains no universally accepted

method for evaluating document-level translation quality. Developing more robust and comprehensive evaluation strategies remains an important open problem, complementary to ours, which we leave for future work.

## 3.2 Results and Analysis

**DocMT-LLMs achieve superior document-to-document (Doc2Doc) translation quality.** DocMT-LLMs (LLM-based document-level MT models) consistently outperform sentence-level models across all benchmarks. On both GuoFeng (Table 2) and IWSLT2017 (Table 3) datasets, DocMT-LLMs all outperform their sentence-level counterparts, with strong gains across both sentence- and document-level metrics. These lower scores for sentence-level baselines are expected, as they lack access to document-wide context during training, which is crucial for ensuring translation coherence. Despite being much smaller, our models outperform strong baselines like LLAMA-3.3-70B-INSTRUCT, QWEN2.5-72B-INSTRUCT and GPT-4O on GuoFeng and perform well on IWSLT2017. While these baselines slightly lead on d-COMET and on d-BLEU (only for IWSLT2017 en→xx), our models still achieve substantial gains over undertrained sentence-level LLMs. These results demonstrate that incorporating DOCBLOCKS enables sentence-level LLMs to generate coherent, high-quality document-level translations.

| Models | GuoFeng zh→en | | | | |
|---|---|---|---|---|---|
| | BLEU | COMET | d-BLEU (BP) | d-COMET | LTCR |
| Llama-3.3-70B-Instruct | 12.12 | 80.16 | 15.32 (0.83) | 71.37 | 65.24 |
| Qwen2.5-72B-Instruct | 17.87 | **84.85** | 20.99 (0.97) | 72.13 | 65.72 |
| GPT-4o | 14.61 | 73.66 | 17.46 (1.00) | **73.73** | 59.69 |
| TowerInstruct-Mistral-7B | 6.78 | 58.40 | 7.27 (1.00) | 58.23 | 54.16 |
| DocMT-TowerInstruct-Mistral-7B | **34.06** | 78.72 | **37.57 (1.00)** | 72.84 | **74.59** |
| EuroLLM-9B-Instruct | 12.43 | 63.14 | 12.03 (0.84) | 60.11 | 51.07 |
| DocMT-EuroLLM-9B-Instruct | 24.12 | 73.88 | 27.83 (1.00) | 72.41 | 67.05 |
| Qwen2.5-7B-Instruct | 16.51 | **84.34** | 19.56 (0.96) | 71.66 | 63.22 |
| DocMT-Qwen2.5-7B-Instruct | 30.67 | 77.28 | 34.80 (0.99) | 71.88 | 73.48 |

Table 2: Doc2Doc translation results on the GuoFeng dataset. Best overall values are **bolded**, and group-specific bests are underlined.

| Models | IWSLT2017 en→xx | | | | | IWSLT2017 xx→en | | | | |
|---|---|---|---|---|---|---|---|---|---|---|
| | BLEU | COMET | d-BLEU (BP) | d-COMET | LTCR | BLEU | COMET | d-BLEU (BP) | d-COMET | LTCR |
| Llama-3.3-70B-Instruct | 15.61 | 61.17 | 27.08 (0.85) | 70.56 | 57.64 | 12.34 | 54.87 | 37.91 (0.90) | 72.01 | 82.20 |
| Qwen2.5-72B-Instruct | 18.48 | 68.47 | 26.59 (0.93) | **75.15** | 60.75 | 13.12 | 58.85 | 38.81 (0.93) | 68.44 | 81.57 |
| GPT-4o | **27.96** | **74.79** | **32.25 (0.97)** | 74.73 | 59.05 | **28.53** | **76.55** | 33.47 (0.89) | **75.04** | 77.88 |
| TowerInstruct-Mistral-7B | 1.77 | 41.67 | 2.41 (0.62) | 29.68 | 27.53 | 10.10 | 59.64 | 2.36 (0.21) | 31.79 | 28.88 |
| DocMT-TowerInstruct-Mistral-7B | 22.67 | 77.33 | 25.60 (0.94) | 74.38 | 60.46 | 27.73 | 59.69 | 26.79 (1.00) | 60.29 | 79.97 |
| EuroLLM-9B-Instruct | 6.51 | 73.64 | 3.75 (0.56) | 24.97 | 35.20 | 8.83 | 78.94 | 3.97 (0.39) | 29.81 | 33.79 |
| DocMT-EuroLLM-9B-Instruct | 20.94 | 72.73 | 20.24 (1.00) | 61.74 | 55.47 | 28.75 | 69.66 | 28.46 (1.00) | 62.71 | 78.20 |
| Qwen2.5-7B-Instruct | 17.32 | 70.57 | 20.86 (0.80) | 71.73 | 56.36 | 23.94 | 75.95 | 34.59 (0.90) | 73.29 | 80.50 |
| DocMT-Qwen2.5-7B-Instruct | 23.98 | **78.48** | 25.29 (0.98) | 73.00 | **62.05** | **37.64** | **81.91** | **40.56 (0.97)** | 73.02 | **84.09** |

Table 3: Doc2Doc translation results on IWSLT2017, averaged over en→xx and xx→en. Best overall values are **bolded**, and group-specific bests are underlined.

**DocMT-LLMs deliver high-quality chunking decoding.** Chunking-based decoding enables efficient document translation by processing text in segments. Larger chunks provide more context, bridging sentence- and document-level translation. DocMT-LLMs consistently outperform sentence-level models on d-BLEU and d-COMET (Figure 3), especially at larger chunk sizes. In contrast, TOWERINSTRUCT-MISTRAL-7B and EUROLLM-9B-INSTRUCT degrade as the chunk size increases. Document-level models maintain strong performance and often surpass QWEN2.5-72B-INSTRUCT. DOCMT-QWEN2.5-7B-INSTRUCT improves quality across most chunk sizes, despite QWEN2.5-7B-INSTRUCT already supporting document-level MT. These results show that training on DOCBLOCKS enhances both document translation and chunking decoding, making DocMT-LLMs more robust.

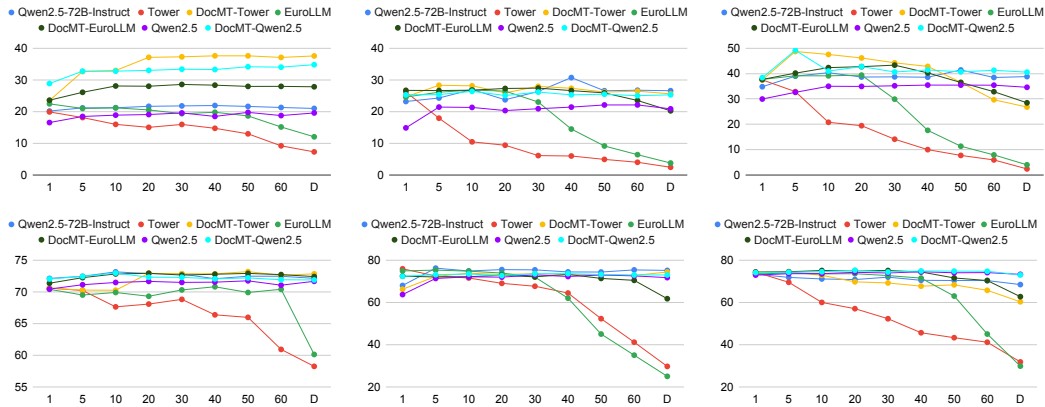

Figure 3: d-BLEU (top) and d-COMET (bottom) scores for the GuoFeng (left), IWSLT2017 en→xx (middle), and xx→en (right) datasets as the decoding chunk size increases. "D" indicates that the entire document is treated as a single chunk during decoding.

**DocMT-LLMs can be coupled with contextual chunking and quality-aware chunking to improve translation quality.** DocMT-LLMs improve translation quality by incorporating context during chunk-level decoding. Figure 4 shows that context-aware prompt tuning (CAPT) helps models better use surrounding context, leading to higher scores across document-level metrics. Quality-aware chunking (Figure 5) further improves performance by combining contextual chunking with paragraph-level metrics like COMET and CON-TEXTCOMET, using a context window of three chunks within a 512-token limit. However, such metrics may not fully capture document-level quality and can lead to suboptimal optimization. As shown in Table 4, standard chunking is the most efficient due to fully parallelizable decoding. In contrast, contextual and quality-aware chunking require sequential decoding, with quality-aware chunking being slower due to candidate reranking with quality metrics. Nevertheless, DOCMT-TOWERINSTRUCT-MISTRAL-7B supports multiple translation paradigms, offering varying translation quality and efficiency trade-offs.

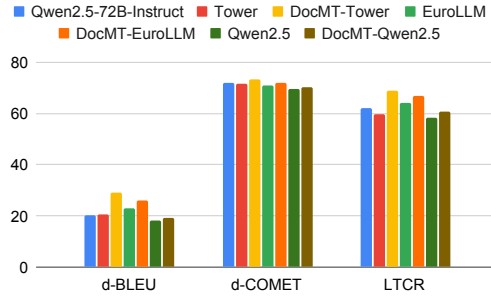

Figure 4: Evaluation scores for contextual chunking on GuoFeng using a context window of 3, consistent with the training setup. Improvements demonstrate CAPT's effect in enabling DocMT-LLMs to better leverage prior translated chunks during decoding.

Figure 5: DOCMT-TOWERINSTRUCT-7B performance on GuoFeng using chunking (C), contextual chunking (CC), and quality-aware chunking (QAC), which combines contextual chunking with paragraph-level quality metrics.

| Inference Method | Chunking | Contextual Chunking | Quality-aware Chunking | Document-to-Document |
|---|---|---|---|---|
| **Throughput ↑** | 392.45 | 39.53 | 16.67 | 204.22 |

Table 4: Throughput comparison of inference methods on Guofeng with DOCMT-TOWERINSTRUCT-MISTRAL-7B, measured in tokens per second.

**Sentence-level MT capabilities of DocMT-LLMs do not deteriorate despite document-level MT training on DOCBLOCKS.** Table 5 shows that DocMT-LLMs fine-tuned with DOCBLOCKS retain strong sentence-level performance, with COMET score differences from their sentence-level counterparts typically within 0.5 points, indicating no catastrophic forgetting. QWEN2.5-7B-INSTRUCT even improves sentence-level performance through document-level training, enhancing robustness without sacrificing sentence-level quality. Detailed results per language pair are provided in Appendix B.

| Models | FLORES-200 | | WMT23 | | TICO-19 |
|---|---|---|---|---|---|
| | en→xx | xx→en | en→xx | xx→en | en→xx |
| TowerInstruct-Mistral-7B | 88.96 | 88.45 | 85.26 | 83.03 | 87.46 |
| DocMT-TowerInstruct-Mistral-7B | 88.54 | 88.21 | 84.93 | 82.97 | 87.17 |
| EuroLLM-9B-Instruct | 89.21 | 88.31 | 85.54 | 83.24 | 87.53 |
| DocMT-EuroLLM-9B-Instruct | 88.79 | 88.41 | 85.26 | 82.88 | 87.20 |
| Qwen2.5-7B-Instruct | 83.08 | 86.70 | 79.89 | 79.81 | 83.36 |
| DocMT-Qwen2.5-7B-Instruct | 87.77 | 88.06 | 83.91 | 81.80 | 86.51 |

Table 5: Evaluation of sentence-level LLMs and DocMT-LLMs based on COMET across sentence-level MT benchmarks.

**Comparing translation approaches: DocMT-LLMs vs. Agent-based methods.** DocMT-LLMs consistently outperform (in quality and inference speed) agent-based methods like TRANSAGENTS (Wu et al., 2024b) and DELTA (Wang et al., 2024) in both Doc2Doc and contextual chunking translation tasks (Figure 6). Agent-based methods iteratively refine translations using memory-based mechanisms with dictionaries and using a hierarchical workflow. This multi-step pipeline and inter-agent communication can introduce significant latency, slowing inference. In contrast, training a DocMT-LLM requires 120 hours on 4 NVIDIA H100 80GB GPUs but runs efficiently on a single NVIDIA A6000 48GB GPU afterward. This initial cost is offset after processing just 42 documents of 400 sentences each. Therefore, both contextual chunking and Doc2Doc approaches with DocMT-LLMs achieve a superior balance among context preservation, speed, scalability, and translation quality.

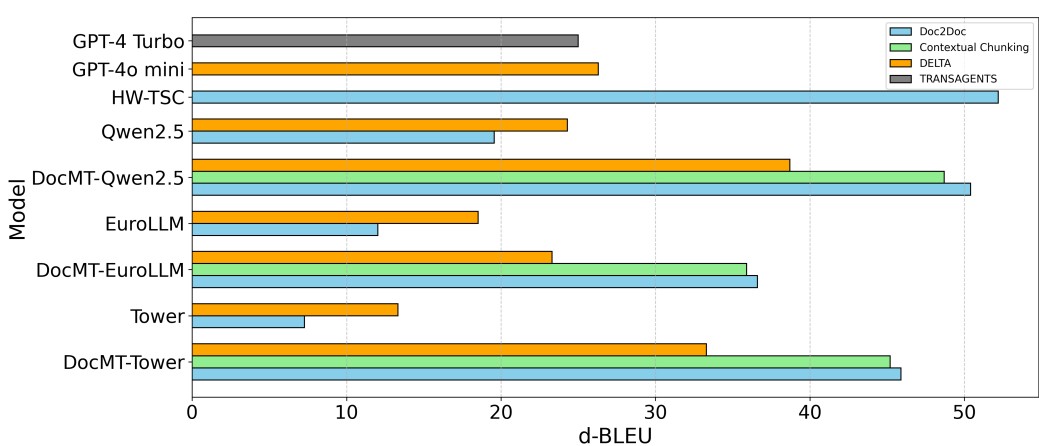

Figure 6: d-BLEU scores on the Guofeng test set, with chapter translations concatenated into a single document (Wu et al., 2024b). The figure compares document-level translation quality across different paradigms: blue for Doc2Doc, green for Contextual Chunking, orange for the DELTA agent-based approach, and gray for the TRANSAGENTS agent-based approach. In addition to evaluating our base models before and after DocMT training, we include strong external baselines representative of each paradigm – TRANSAGENTS with GPT-4 Turbo (Wu et al., 2024b), DELTA with GPT-4o mini (Wang et al., 2024), and HW-TSC (Xie et al., 2023) for Doc2Doc.

**DOCBLOCKS curation process impacts learning.** As shown in Table 6, filtering improves model performance across most datasets. The Multiresolution Document-to-Document (MRD2D) technique provides significant benefits for both document-to-document translation and chunking by introducing additional subdocuments during training, especially for the IWSLT2017 dataset. Context-Aware Prompt Tuning (CAPT) is particularly effective for Contextual Chunking, as it provides context-driven examples that improve performance, particularly for the GuoFeng and IWSLT2017 (xx→en) tasks. Combining filtering with MRD2D and CAPT yields the best results overall, boosting translation quality and contextual understanding across document-level translation tasks. We further examine the impact of balancing the DOCBLOCKS dataset, which is naturally skewed toward English-to-Chinese document pairs due to their greater availability. In this setting, we uniformly sample both from and into English across language directions to ensure an even distribution of translation pairs. Our results indicate that enforcing this balance yields no performance gains. As such, we choose to retain as much data as possible rather than discard samples to achieve balance, prioritizing overall model performance.

| Models | Doc2Doc | | | Chunking | | | Contextual Chunking | | |
|---|---|---|---|---|---|---|---|---|---|
| | GuoFeng | IWSLT2017 | | GuoFeng | IWSLT2017 | | GuoFeng | IWSLT2017 | |
| | zh→en | en→xx | xx→en | zh→en | en→xx | xx→en | zh→en | en→xx | xx→en |
| TOWERINSTRUCT-MISTRAL-7B | 58.23 | 29.68 | 31.79 | 68.81 | 67.61 | 52.26 | 70.21 | 70.62 | 69.53 |
| **SFT** | | | | | | | | | |
| + unfiltered DOCBLOCKS | 70.08 | 62.30 | 55.32 | 69.20 | 69.24 | 55.89 | 70.55 | 69.34 | 69.00 |
| + filtered DOCBLOCKS | 71.71 | 63.67 | 57.72 | 69.21 | 70.36 | 58.23 | 70.45 | 69.77 | 68.44 |
|   + MRD2D | 72.55 | **74.60** | 60.11 | 72.38 | 73.03 | 68.97 | 70.34 | 71.32 | 69.45 |
|   + CAPT | 71.70 | 66.33 | 58.31 | 68.80 | 68.55 | 65.37 | **73.11** | 73.05 | **75.04** |
|   + MRD2D + CAPT | **72.84** | 74.38 | **60.29** | **72.86** | **73.51** | **69.21** | 72.93 | **73.18** | 74.48 |
| + balanced DOCBLOCKS | 67.89 | 61.79 | 55.03 | 65.72 | 69.57 | 54.39 | 69.80 | 70.33 | 68.90 |

Table 6: Ablation results for the components of DOCBLOCKS, reporting d-COMET scores.

**Balancing sentence- and document-Level data in DOCBLOCKS: impacts on translation performance.** We conducted an ablation study to assess how varying the proportion of sentence-level data in DOCBLOCKS affects translation performance, while keeping document-level data constant. The sentence-level portion is sourced from TowerBlocks (Alves et al., 2024) and uniformly sampled across languages. Results shown in Figure 7 demonstrate that just 10% sentence-level data is sufficient to maintain sentence-level quality, while higher proportions yield diminishing returns and reduce document-level performance. Allocating 90% of training data to document-level pairs provides the best balance, enhancing document-level performance without compromising sentence-level capabilities.

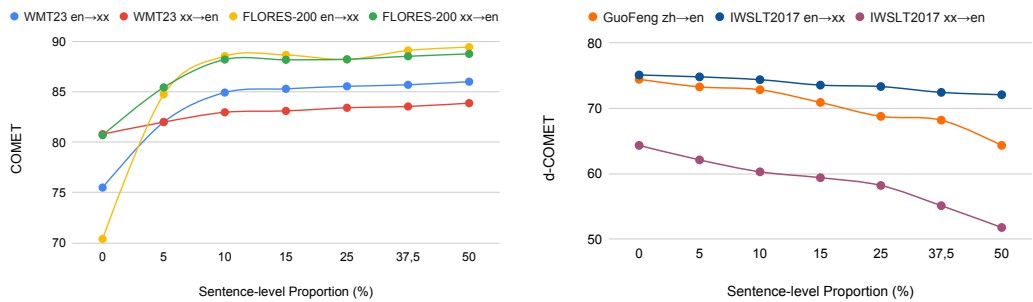

Figure 7: Effect of varying sentence-level data in DOCBLOCKS training on sentence-level (left) and document-level (right) translation performance.

# 4 Related Work

**Document-Level Machine Translation.** Traditional sentence-level MT (Sutskever et al., 2014; Bahdanau et al., 2014; Vaswani et al., 2017) often leads to inconsistencies in terminology, tone, and context in longer texts (Xiong et al., 2018; Hu & Wan, 2023; Wang et al., 2023b). Document-level MT (Maruf et al., 2021) improves coherence by incorporating broader context across sentences and paragraphs. Recent approaches include document embeddings (Macé & Servan, 2019; Huo et al., 2020), multi-encoder architectures (Zhang et al., 2018; Voita et al., 2018; Bawden et al., 2018), and enhanced attention mechanisms for long-range dependencies (Zhang et al., 2020; Miculicich et al., 2018; Wu et al., 2023). While promising, progress is limited by the lack of high-quality document-level parallel data (Liu & Zhang, 2020; Wang et al., 2023b), which hinders training in complex domains such as literary translation (Ghazvininejad et al., 2018; Toral & Way, 2018). Recent efforts have drawn on web crawls (e.g., ParaCrawl (Bañón et al., 2020), BWB (Jiang et al., 2022)), news data (e.g., WMT (Kocmi et al., 2024)), parliamentary records (e.g., Europarl (Koehn, 2005)), public talks (e.g., IWSLT TED task (Cettolo et al., 2017)), literary texts (e.g., GuoFeng (Wang et al., 2023b)), and subtitles (e.g., OpenSubtitles (Lison & Tiedemann, 2016)), as well as large multilingual corpora (e.g., UNPC (Ziemski et al., 2016)). However, many of these suffer from domain constraints, noisy alignments, or unclear document boundaries. We address these issues by curating DOCBLOCKS, a high-quality document-level dataset built from public sources, emphasizing clean alignment, coherent document structure, and careful filtering.

**LLMs for Machine Translation.** LLMs have recently demonstrated strong performance on MT tasks (Alves et al., 2024). Their capacity to process long contexts opens new possibilities for modeling discourse structure and inter-sentence dependencies, which are essential for document-level translation. To leverage these capabilities, new strategies are being explored. These include context-aware prompting, which uses relevant context directly in the prompt to guide translation (Wang et al., 2023a; Wu et al., 2024), and document-level fine-tuning, which adapts instruction-tuned models to achieve stronger performance (Wu et al., 2024; Wu et al., 2024a; Li et al., 2024). In addition, agent-based approaches such as DELTA (Wang et al., 2024) and TRANSAGENTS (Wu et al., 2024b) further improve consistency by maintaining structured memory across sentences, though they introduce substantial inference overhead. Despite these advances, LLMs still face challenges in maintaining coherence over long documents due to limitations in training data (Wu et al., 2024a) and context modeling (Karpinska & Iyyer, 2023). To address these challenges, we fine-tune LLMs using DOCBLOCKS, enhancing their multilingual contextual understanding for document translation without incurring the high cost of full retraining.

# 5 Conclusion

This study advances the transition of state-of-the-art LLMs from sentence-level MT to document-level MT by addressing key challenges in contextual coherence and long-document consistency. Fine-tuning on the curated DOCBLOCKS dataset allows DocMT-LLMs to effectively manage long-range dependencies while preserving sentence-level translation quality. Moreover, context-aware prompt tuning and multi-resolution document-to-document training enhance adaptability across various translation paradigms, including document-to-document and several chunk-level methods. These findings provide a cost-effective and scalable approach to improving document-level translation quality.

# Acknowledgments

We thank the members of the SARDINE lab for their useful and constructive comments. This work was supported by the Portuguese Recovery and Resilience Plan through project C645008882-00000055 (Center for Responsible AI), by the EU's Horizon Europe Research and Innovation Actions (UTTER, contract 101070631), by the project DECOLLAGE (ERC-2022-CoG 101088763), and by Fundação para a Ciência e Tecnologia through contract UIDB/50008/2020.

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

# A   Document-level Training Hyperparameters

Table 7 contains the hyperparameter configuration for the training of DocMT-LLMs.

| | |
|---|---|
| Batch size | 32 |
| Number of Epochs | 2 |
| Learning rate | $7 \times 10^{-6}$ |
| LR Scheduler | cosine |
| Warmup Steps | 125 |
| Weight Decay | 0.01 |
| Optimizer | Adam (Kingma & Ba, 2014) |
| Adam $\beta_1$ | 0.9 |
| Adam $\beta_2$ | 0.999 |
| Adam $\epsilon$ | $1 \times 10^{-8}$ |
| Maximum Sequence Length | 32768 |

Table 7: Hyperparameter configuration to fine-tune DocMT-LLMs on DocBlocks.

# B   Full Translation Results

In this appendix, we provide detailed sentence-level and document-level evaluation results, broken down by individual language pairs. Tables 8, 9, and 10 report document-level evaluation results across IWSLT2017 language pairs using d-BLEU, d-COMET, and LTCR, respectively – similar to the results presented in Table 3. Additionally, Tables 11, 12, and 13 present the same comparisons as Table 5 in the main paper, illustrating the impact of document-level training on sentence-level translation performance across diverse datasets and language directions.

| Models | IWSLT2017 (en→xx) | | | | | | | | |
|---|---|---|---|---|---|---|---|---|---|
| | de | es | fr | it | ko | nl | pt | ru | zh |
| Llama-3.3-70B-Instruct | 30.51 | 35.17 | 34.45 | 28.96 | 19.59 | **32.20** | 25.08 | 15.12 | 22.64 |
| Qwen2.5-72B-Instruct | 24.86 | 34.34 | 34.63 | 28.65 | 20.54 | 30.73 | 24.74 | 13.50 | 27.32 |
| GPT-4o | 30.49 | **50.21** | 33.51 | **30.29** | **22.71** | 30.39 | **42.21** | **35.21** | 15.21 |
| TowerInstruct-Mistral-7B | 2.63 | 2.07 | 3.82 | 2.36 | 0.41 | 3.00 | 4.12 | 1.83 | 1.44 |
| DocMT-TowerInstruct-Mistral-7B | 28.58 | 23.92 | **35.45** | 29.34 | 18.59 | 26.51 | 27.72 | 13.15 | 27.13 |
| EuroLLM-9B-Instruct | 4.42 | 5.53 | 6.65 | 3.45 | 0.33 | 3.29 | 3.75 | 4.69 | 1.64 |
| DocMT-EuroLLM-9B-Instruct | 21.21 | 32.19 | 19.91 | 15.98 | 14.30 | 18.98 | 29.62 | 12.49 | 17.48 |
| Qwen2.5-7B-Instruct | 26.51 | 31.88 | 30.20 | **24.80** | 0.27 | **29.28** | 23.35 | **9.84** | 11.62 |
| DocMT-Qwen2.5-7B-Instruct | **31.08** | 33.59 | 34.85 | 23.19 | 19.43 | 27.83 | 20.52 | 8.09 | **29.03** |
| Models | IWSLT (xx→en) | | | | | | | | |
| | de | es | fr | it | ko | nl | pt | ru | zh |
| Llama-3.3-70B-Instruct | 46.02 | **45.55** | 45.56 | **45.21** | 16.63 | **48.87** | **40.60** | **28.75** | 24.01 |
| Qwen2.5-72B-Instruct | **46.07** | 44.11 | **45.66** | 44.86 | **24.86** | 48.52 | 40.06 | 28.20 | 26.96 |
| GPT-4o | 42.36 | 40.36 | 41.36 | 39.36 | 13.36 | 43.36 | 36.36 | 23.36 | 21.36 |
| TowerInstruct-Mistral-7B | 3.15 | 2.49 | 3.57 | 2.94 | 0.78 | 4.08 | 2.97 | 0.95 | 0.32 |
| DocMT-TowerInstruct-Mistral-7B | 18.38 | 31.98 | 23.01 | 38.61 | 30.79 | 19.31 | 27.99 | 17.84 | **33.20** |
| EuroLLM-9B-Instruct | 6.31 | 0.55 | 6.86 | 4.29 | 3.74 | 4.46 | 1.91 | 4.66 | 2.95 |
| DocMT-EuroLLM-9B-Instruct | 31.92 | 33.87 | 21.32 | 22.43 | 28.35 | 37.94 | 30.41 | 18.37 | 31.54 |
| Qwen2.5-7B-Instruct | 40.78 | 40.28 | 41.39 | 41.38 | 20.83 | 41.15 | 36.61 | 24.70 | 24.18 |
| DocMT-Qwen2.5-7B-Instruct | 46.00 | 44.05 | **47.33** | 44.93 | **36.94** | 46.45 | 37.88 | **30.35** | 31.11 |

Table 8: Evaluation based on d-BLEU across IWSLT2017 language pairs.

| Models | IWSLT2017 (en→xx) | | | | | | | | |
|---|---|---|---|---|---|---|---|---|---|
| | de | es | fr | it | ko | nl | pt | ru | zh |
| Llama-3.3-70B-Instruct | 67.90 | 71.31 | 66.91 | 69.81 | 61.11 | 72.49 | 70.33 | 74.00 | 81.18 |
| Qwen2.5-72B-Instruct | 71.26 | **73.16** | 69.50 | 72.49 | **81.45** | 74.55 | 72.03 | **77.82** | 84.08 |
| GPT-4o | 74.90 | 72.38 | 64.67 | 75.39 | 80.69 | 76.49 | 69.38 | 74.38 | **84.30** |
| TowerInstruct-Mistral-7B | 27.78 | 28.63 | 30.04 | 27.18 | 30.57 | 27.29 | 30.41 | 30.72 | 34.50 |
| DocMT-TowerInstruct-Mistral-7B | **76.39** | 63.74 | **72.47** | **77.51** | 74.62 | **79.13** | **74.04** | 75.88 | 75.64 |
| EuroLLM-9B-Instruct | 20.70 | 30.50 | 20.49 | 24.24 | 20.72 | 23.78 | 29.04 | 33.26 | 21.99 |
| DocMT-EuroLLM-9B-Instruct | 64.01 | 66.37 | 58.64 | 63.91 | 57.12 | 67.94 | 68.34 | 69.65 | 39.68 |
| Qwen2.5-7B-Instruct | 68.92 | 71.42 | 66.51 | 68.39 | 76.40 | 69.91 | 69.49 | 71.45 | 83.08 |
| DocMT-Qwen2.5-7B-Instruct | 71.13 | 72.75 | 68.55 | 72.59 | 77.39 | 75.15 | 64.40 | 71.84 | 83.21 |

| Models | IWSLT2017 (xx→en) | | | | | | | | |
|---|---|---|---|---|---|---|---|---|---|
| | de | es | fr | it | ko | nl | pt | ru | zh |
| Llama-3.3-70B-Instruct | 71.14 | 72.82 | 72.24 | 69.97 | 78.11 | 67.30 | 69.29 | 73.97 | 73.24 |
| Qwen2.5-72B-Instruct | 68.95 | 70.49 | 65.72 | 62.37 | 70.45 | 64.92 | 71.12 | 70.39 | 71.55 |
| GPT-4o | **75.74** | **73.45** | 66.79 | **76.45** | 78.95 | **77.25** | 67.89 | **75.60** | **83.25** |
| TowerInstruct-Mistral-7B | 34.08 | 32.01 | 30.77 | 33.88 | 30.93 | 33.34 | 31.87 | 24.63 | 34.59 |
| DocMT-TowerInstruct-Mistral-7B | 55.78 | 68.36 | 51.34 | 64.15 | 54.80 | 54.30 | 69.13 | 68.47 | 56.30 |
| EuroLLM-9B-Instruct | 29.38 | 30.10 | 28.25 | 29.48 | 29.29 | 27.55 | 32.17 | 30.90 | 31.15 |
| DocMT-EuroLLM-9B-Instruct | 63.62 | 70.64 | 53.36 | 68.65 | 44.61 | 67.55 | 72.65 | 71.65 | 51.67 |
| Qwen2.5-7B-Instruct | 72.54 | 71.90 | 71.66 | 73.81 | 77.42 | 74.23 | 73.40 | 73.47 | 71.17 |
| DocMT-Qwen2.5-7B-Instruct | 74.63 | 63.36 | **73.28** | 74.80 | **79.14** | 75.39 | **74.33** | 72.64 | 69.60 |

Table 9: Evaluation based on d-COMET across IWSLT2017 language pairs.

| Models | IWSLT2017 (en→xx) | | | | | | | | |
|---|---|---|---|---|---|---|---|---|---|
| | de | es | fr | it | ko | nl | pt | ru | zh |
| Llama-3.3-70B-Instruct | 60.37 | 76.33 | 61.01 | 64.43 | 24.09 | 58.58 | 62.65 | **56.77** | 54.52 |
| Qwen2.5-72B-Instruct | 62.41 | **79.40** | 61.25 | 66.03 | 41.19 | 59.35 | 65.65 | 56.23 | **55.24** |
| GPT-4o | **71.83** | 61.97 | 60.32 | 65.66 | **56.01** | 61.75 | 57.97 | 48.97 | 46.97 |
| TowerInstruct-Mistral-7B | 25.96 | 26.55 | 24.68 | 44.73 | 26.17 | 27.10 | 29.06 | 19.27 | 24.24 |
| DocMT-TowerInstruct-Mistral-7B | 59.93 | 72.67 | **64.31** | 63.59 | 41.62 | 66.95 | **68.64** | 55.77 | 50.66 |
| EuroLLM-9B-Instruct | 45.05 | 33.63 | 48.45 | 48.25 | 26.78 | 37.82 | 26.75 | 17.58 | 32.48 |
| DocMT-EuroLLM-9B-Instruct | 56.79 | 73.23 | 60.23 | 62.60 | 31.65 | 47.98 | 68.51 | 48.13 | 50.11 |
| Qwen2.5-7B-Instruct | 57.74 | 70.77 | 59.10 | 61.28 | 37.16 | 54.79 | 65.87 | 47.87 | 52.66 |
| DocMT-Qwen2.5-7B-Instruct | 61.21 | 76.78 | 62.86 | **71.58** | 45.22 | **67.71** | 68.25 | 50.32 | 54.51 |

| Models | IWSLT (xx→en) | | | | | | | | |
|---|---|---|---|---|---|---|---|---|---|
| | de | es | fr | it | ko | nl | pt | ru | zh |
| Llama-3.3-70B-Instruct | 84.69 | 88.38 | 88.54 | 88.60 | 69.32 | **88.49** | 81.22 | 77.56 | 73.02 |
| Qwen2.5-72B-Instruct | 83.00 | 87.61 | 87.90 | 87.80 | 71.46 | 85.79 | **81.35** | 77.60 | 71.63 |
| GPT-4o | 82.07 | 83.98 | 82.79 | 85.40 | 65.81 | 79.92 | 77.23 | 74.94 | 68.75 |
| TowerInstruct-Mistral-7B | 30.12 | 32.42 | 27.00 | 38.22 | 27.69 | 29.69 | 21.60 | 24.88 | 28.29 |
| DocMT-TowerInstruct-Mistral-7B | 84.28 | **90.10** | 87.97 | **91.87** | 60.02 | 82.90 | 80.16 | **79.26** | 63.16 |
| EuroLLM-9B-Instruct | 39.17 | 34.80 | 40.49 | 35.17 | 26.13 | 56.22 | 28.75 | 20.54 | 22.83 |
| DocMT-EuroLLM-9B-Instruct | 77.73 | 83.92 | 81.75 | 86.86 | 69.93 | 79.49 | 80.07 | 75.73 | 68.33 |
| Qwen2.5-7B-Instruct | 82.55 | 83.48 | 89.01 | 88.71 | 68.41 | 86.08 | 79.33 | 75.44 | 71.49 |
| DocMT-Qwen2.5-7B-Instruct | **85.65** | 84.49 | **91.16** | 91.72 | **74.74** | 86.82 | 80.88 | 76.26 | **85.09** |

Table 10: Evaluation based on LTCR across IWSLT2017 language pairs.

| Models | WMT23 | | | | | |
|---|---|---|---|---|---|---|
| | en→de | en→ru | en→zh | de→en | ru→en | zh→en |
| TowerInstruct-Mistral-7B | 83.91 | 85.77 | 86.10 | 85.48 | 83.15 | 80.46 |
| DocMT-TowerInstruct-Mistral-7B | 83.70 | 85.47 | 85.62 | 85.39 | 82.93 | 80.59 |
| EuroLLM-9B-Instruct | 84.83 | 85.64 | 86.15 | 85.48 | 83.34 | 80.90 |
| DocMT-EuroLLM-9B-Instruct | 84.55 | 85.62 | 85.61 | 85.31 | 82.72 | 80.61 |
| Qwen2.5-7B-Instruct | 78.45 | 76.78 | 84.45 | 81.92 | 78.92 | 78.59 |
| DocMT-Qwen2.5-7B-Instruct | 81.86 | 84.21 | 85.66 | 83.44 | 81.73 | 80.23 |

Table 11: Evaluation based on COMET across WMT23 language pairs.

| Models | TICO-19 | | | | |
|---|---|---|---|---|---|
| | en→es | en→fr | en→pt | en→ru | en→zh |
| TowerInstruct-Mistral-7B | 88.60 | 82.03 | 89.51 | 88.31 | 88.85 |
| DocMT-TowerInstruct-Mistral-7B | 88.73 | 81.72 | 89.55 | 88.16 | 87.70 |
| EuroLLM-9B-Instruct | 88.61 | 81.82 | 90.01 | 88.48 | 88.74 |
| DocMT-EuroLLM-9B-Instruct | 88.96 | 81.94 | 89.49 | 88.82 | 86.79 |
| Qwen2.5-7B-Instruct | 85.51 | 78.17 | 87.56 | 78.77 | 86.79 |
| DocMT-Qwen2.5-7B-Instruct | 88.04 | 81.03 | 88.99 | 87.12 | 87.37 |

Table 12: Evaluation based on COMET across TICO-19 language pairs.

| Models | FLORES-200 (en→xx) | | | | | | | | |
|---|---|---|---|---|---|---|---|---|---|
| | de | es | fr | it | ko | nl | pt | ru | zh |
| TowerInstruct-Mistral-7B | 88.25 | 87.05 | 88.89 | 89.20 | 90.11 | 88.67 | 89.77 | 90.13 | 88.56 |
| DocMT-TowerInstruct-Mistral-7B | 88.52 | 86.79 | 88.55 | 88.75 | 89.20 | 88.59 | 89.68 | 89.76 | 87.01 |
| EuroLLM-9B-Instruct | 88.92 | 87.38 | 89.10 | 89.29 | 89.96 | 88.68 | 90.39 | 90.38 | 88.79 |
| DocMT-EuroLLM-9B-Instruct | 88.71 | 86.94 | 88.74 | 88.94 | 89.81 | 88.59 | 89.39 | 90.26 | 87.73 |
| Qwen2.5-7B-Instruct | 83.21 | 84.10 | 84.92 | 83.77 | 77.81 | 80.24 | 87.04 | 79.40 | 87.21 |
| DocMT-Qwen2.5-7B-Instruct | 86.97 | 86.37 | 87.81 | 87.71 | 88.39 | 86.53 | 88.93 | 88.94 | 88.28 |
| Models | FLORES-200 (xx→en) | | | | | | | | |
| | de | es | fr | it | ko | nl | pt | ru | zh |
| TowerInstruct-Mistral-7B | 89.50 | 87.62 | 89.55 | 88.50 | 88.50 | 87.95 | 90.07 | 87.09 | 87.25 |
| DocMT-TowerInstruct-Mistral-7B | 89.47 | 87.24 | 89.44 | 88.20 | 88.30 | 87.67 | 89.78 | 86.84 | 86.95 |
| EuroLLM-9B-Instruct | 89.41 | 87.25 | 89.59 | 88.29 | 88.54 | 87.59 | 89.75 | 86.93 | 87.44 |
| DocMT-EuroLLM-9B-Instruct | 89.65 | 87.77 | 89.60 | 88.39 | 88.35 | 87.57 | 90.02 | 87.17 | 87.17 |
| Qwen2.5-7B-Instruct | 87.98 | 86.24 | 88.37 | 86.82 | 85.73 | 85.59 | 87.92 | 85.37 | 86.28 |
| DocMT-Qwen2.5-7B-Instruct | 89.25 | 87.38 | 89.20 | 88.04 | 87.74 | 87.41 | 89.58 | 86.77 | 87.16 |

Table 13: Evaluation based on COMET across FLORES-200 language pairs.

