# OpenReview forum: "Multilingual Contextualization of Large Language Models for Document-Level Machine Translation"
_colmweb.org/COLM/2025/Conference — COLM 2025_

### Official Review · Reviewer_eZn7 · 2025-05-10

**Rating:** 6
**Confidence:** 4
**Ethics Flag:** 1

**Summary:**

This paper introduces a new document-level translation dataset, DOCBLOCKS, which contains document-level translation pairs across multiple language pairs. DOCBLOCKS is constructed from several publicly available document-level datasets. The authors then fine-tune LLMs using this dataset through two methods: document-to-document (doc2doc) translation and context-aware translation. A comprehensive evaluation is conducted, demonstrating the effectiveness of their approach in both doc2doc and context-aware translation tasks.

**Questions To Authors:**

1. According to Figure 2, en<->zh document pairs make up a significant portion of DOCBLOCKS. How does this imbalance affect the performance on other language pairs? Does it help or hinder their translation quality?

2. Do the source and target documents in DOCBLOCKS have the same number of sentences? Since real-world document translations often vary in sentence alignment, it would be helpful to see a discussion on how this is handled.

**Reasons To Accept:**

1. The paper presents DOCBLOCKS, a potentially valuable resource for fine-tuning LLMs for document-level translation across various language pairs.

2. The evaluation is thorough, showcasing the model's translation performance in document-level translation.

**Reasons To Reject:**

1. Since DOCBLOCKS is entirely constructed from publicly available datasets, the novelty and contribution of the dataset itself may be limited. Figure 2 presents statistics for the original datasets used to build DOCBLOCKS, but does not provide detailed statistics for the final DOCBLOCKS dataset. It remains unclear how many documents or sentences were retained after preprocessing.

2. The fine-tuning approach lacks novelty, as it relies on two existing training strategies: MRD2D and CAPT.

---

> ### Author Response · Authors · 2025-06-02
>
> We thank the reviewer for the constructive feedback.
>
> > Since DocBlocks is entirely constructed from publicly available datasets, the novelty and contribution of the dataset itself may be limited.
>
> We respectfully disagree with the assertion that DocBlocks lacks novelty due to its reliance on publicly available corpora. While the raw documents are indeed obtained from public sources, the novelty lies in our rigorous, large-scale curation and filtering process and the purpose-driven design for long-context, multilingual document-level MT -- a resource that, to our knowledge, does not currently exist in comparable scope or quality.
>
> Specifically, DocBlocks is not a simple aggregation of existing datasets. We systematically clean, deduplicate, and segment documents using tools and metrics such as Bicleaner, langid and CometKiwi, applying strict thresholds to filter out noisy or misaligned pairs—where the source and target texts are incorrectly matched (see Section 2.1). This process significantly improves data quality and coherence at the document level, addressing common issues in existing datasets. As recent work shows [1, 2], data quality now outweighs quantity in LLM training. Careful filtering, rather than broad collection, is essential for building effective models, as cleaner datasets can outperform much larger, noisier ones.
>
> DocBlocks also encompasses a wide range of genres and discourse types—including news articles, TED talks, parliamentary proceedings, and novels—enabling broader domain generalization than most prior document-level MT corpora, which are often limited in scope or language coverage.
>
> Furthermore, we apply structured document context transformation techniques, MRD2D and CAPT, to generate more diverse examples that encompass multiple translation paradigms. We show that these transformations, when used together, play a central role in improving downstream model performance.
>
> > Figure 2 presents statistics for the original datasets... but not for the final DocBlocks dataset.
>
> We appreciate the reviewer’s feedback and will include comprehensive statistics such as post-filtering document counts, language pair distributions, and average sentence and document lengths in the final version of the paper. We will also publicly release DocBlocks to facilitate reproducibility and encourage further innovation. Additionally, we emphasize that the impact of our data curation process is quantitatively evaluated in the ablation studies presented in Table 6, which directly link filtering and the use of MRD2D and CAPT techniques to downstream model performance, demonstrating the effectiveness of our curation methodology.
>
> > The fine-tuning approach lacks novelty, as it relies on two existing training strategies: MRD2D and CAPT.
>
> We appreciate the reviewer’s feedback. While it is true that we do not introduce a new fine-tuning algorithm (e.g., a novel loss function), we disagree that our approach lacks novelty. Our contribution lies in designing a practical and effective recipe for adapting sentence-level MT LLMs to document-level translation by integrating existing techniques in a novel and impactful way, and in empirically demonstrating the superiority of this approach over other popular strategies, such as agent-based methods.
>
> Specifically:
> - We show that instruction-tuned sentence-level LLMs can be effectively transformed into strong document-level models through targeted fine-tuning on DocBlocks, using relatively lightweight training. As noted in Line 246, our document-level models are trained on 4 H100 GPUs for approximately 120 hours.
> - We combine MRD2D and CAPT with high-quality, context-rich instructions, leveraging DocBlocks to support multiple document translation paradigms. The effectiveness of this combination and its impact across both Doc2Doc and chunking-based decoding strategies have not been demonstrated in prior work. Furthermore, we show that applying these techniques together results in consistent improvements across all decoding methods (see Table 6), which, to our knowledge, has not been previously explored or reported.
> - Our resulting models are competitive with, or outperform, significantly larger systems (e.g., GPT-4o) despite using fewer resources.
>
> These contributions, supported by rigorous evaluation across multiple translation paradigms represent a meaningful step toward scalable, open-source, document-level machine translation and offer a practical framework for the community to adopt and extend.

---

> > ### Author Response · Authors · 2025-06-02
> >
> > > According to Figure 2, en<->zh document pairs make up a significant portion of DocBlocks. How does this imbalance affect the performance on other language pairs? Does it help or hinder their translation quality?
> >
> > We thank the reviewer for raising this important point. The prominence of en↔zh in DocBlocks reflects the availability of high-quality, long-form parallel corpora for this language pair—particularly in domains like literature, where document-level translation benefits most from rich discourse context. GuoFeng and BWB are among the few resources that offer full-document translations at this scale and quality.
> >
> > That said, DocBlocks is carefully designed to support multilingual document-level translation, and our models perform well across all language pairs, as evidenced in Table 3.
> >
> > To better quantify the impact of this language distribution, we appreciate the suggestion and will include a detailed analysis in the final version (similar in style to the ablation in Table 5) by downsampling en↔zh and evaluating its effect on other language pairs. This will help clarify the extent of any cross-lingual transfer within DocBlocks.
> >
> > > Do source and target documents have the same number of sentences? [...]
> >
> > In real-world corpora, sentence alignment is often imperfect. During DocBlocks construction, we preserve document-level granularity and perform alignment at the document level, not on a strict sentence-to-sentence basis. This reflects the realistic nature of document translation and improves robustness. For training, we use chunk-level and full-document examples, allowing models to handle variations in alignment and rely more on context and semantics than on rigid 1-to-1 mappings.
> >
> > As explained in the experimental setup (Line 179-182), sentence-level alignment is used only during evaluation to compute sentence-level scores, such as BLEU and COMET via bleualign. In all other stages, including dataset construction, model fine-tuning, and evaluation using document-level metrics, we do not require source and target documents to have the same number of sentences.
> > We will elaborate on this further in the final version for clarity.
> >
> > We hope that our answer alleviates your main concerns. We are happy to address any other question you may have.
> >
> > [1] https://arxiv.org/abs/2309.04564
> >
> > [2] https://aclanthology.org/2023.wmt-1.50

---

> > > ### Comment · Reviewer_eZn7 · 2025-06-10
> > >
> > > Thank you for your explanations. I have raised my rating from 5 to 6.

---

### Official Review · Reviewer_BHRV · 2025-05-10

**Rating:** 6
**Confidence:** 3
**Ethics Flag:** 1

**Summary:**

This paper describes an effort at improving document-level machine translation. It claims three major contributions: creating DocBlocks, a new dataset, and presenting a novel fine-tuning approach, combining sentence and document-level data, and a comprehensive evaluation.

**Questions To Authors:**

In lines 223-24, I don't understand how Figure 4 is supposed to show the advantage of CAPT (which is not even mentioned in its caption), or how it relates to other results.

I donät understand Figure 6. What are all the colors supposed to illustrate? Different variants of the same model? If so, how are your variant models trained?

**Reasons To Accept:**

The approach to document-level fine-tuning is interesting and leads to good results

The combination and cleaning of existing corpora may be useful to others

**Reasons To Reject:**

The evaluation is mainly based on sentence-level metrics, or poor document-level metrics (the latter is acknowledged by the author and left for future work). I understand that evaluation is challenging, but in order to claim stronger document-level performance, with poor metrics, I would expect to see at least a small human evaluation to complement the metrics.

The creation of a new dataset is overstated. I read the claim as a completely new dataset, but it is actually a compilation of many existing datasets (many of which have been used for document-level decoding for over a decade). The main contribution is combining the datasets and cleaning them in a consistent way. This needs to be made much more clear. Clear statistics of the dataset size after cleaning are currently missing. The paper would also be much stronger if you included a comparison using the original datasets (or a random sample of the same size as DB) and compared the performance with DocBlocks' performance.

It is not clearly stated which languages you work on, and the font in Figure 2 is so small that I could not even read it on my print-out. This should be clearly stated early in the paper. I would also expect results per language pair, at least in the appendix, and not only aggregated scores (en-xx, xx-en).

The related work section is minimal. Please expand and more clearly relate your work to previous research if the paper is accepted

I think the claim based on Figure 7, that your document training does not hurt sentence-level decoding, is exaggerated and not clearly supported by the results. Especially, IWSLT xx-en is hurt quite a lot at 10%.  Also, the coloring is misleading, with different datasets in the left and right plots having the same colors. Please use different colors for different datasets.

---

> ### Author Response · Authors · 2025-06-02
>
> We sincerely thank the reviewer for the detailed and constructive feedback.
>
> > The evaluation is mainly based on sentence-level metrics, or poor document-level metrics […] I would expect to see at least a small human evaluation.
>
> We fully acknowledge that evaluating document-level MT remains an open and complex challenge, particularly due to the lack of universally accepted automatic metrics. To mitigate this limitation, we employed multiple document-level metrics as described in Section 3.1. These provide complementary perspectives on translation quality, particularly targeting coherence, consistency, and terminology preservation.
>
> While we agree that these metrics are not perfect, we would respectfully argue that labeling them as “poor” may overlook their continued utility. For instance, BLEU, despite its known limitations, has served as the de facto standard for evaluating MT for decades, and its document-level variant (d-BLEU) still offers useful signals, especially when triangulated with metrics like COMET and LTCR. We view these as valuable proxies that help quantify improvements, particularly in the absence of comprehensive human evaluation.
>
> Regarding human evaluation, we agree it could significantly enrich the analysis. However, given the multilingual nature of our work, the length of test documents (e.g., TED Talks), and the fine-grained nature of discourse-level quality, a robust and reliable human evaluation would require professional translators across many languages, which was infeasible within our current resource constraints. Moreover, the subjectivity inherent in human assessments, particularly for phenomena such as literary style preservation or discourse coherence, complicates the design of a universal evaluation framework [1,2,3]. We expect future work to overcome these challenges in the upcoming years through more document-level human annotations and agreed evaluation protocols in WMT campaigns, with subsequent use of these data to improve automatic evaluation.
>
> > The creation of a new dataset is overstated [...] it's actually a compilation. Clear statistics are missing. [...] Comparison with original datasets is needed.
>
> We appreciate the reviewer’s feedback and agree that the framing of our dataset's contribution can be made clearer. While DocBlocks is indeed a compilation of existing corpora, please note that it is not a simple aggregation. As described in Section 2.1, we applied a rigorous cleaning and curation pipeline, including:
> - Filtering using Bicleaner and CometKiwi,
> - Document-level alignment and deduplication,
> - Language identification filtering.
>
> These steps significantly improve the quality of the data compared to the raw sources—this is key in obtaining high-quality and practically useful datasets.
>
> This is substantiated by the comparison that we actually do with the original, unfiltered datasets in Table 6, where models fine-tuned on the unfiltered DocBlocks data perform much worse than those using the curated version. While we already compare performance using the unfiltered dataset in Table 6, we have not included per-source comparisons, as our aim is to construct a high-quality multilingual dataset rather than optimize for any single corpus.
>
> Regarding dataset statistics, please see Figure 2, which presents counts of documents, sentences, and average lengths across domains. However, we agree that providing additional post-cleaning statistics (e.g., final document counts per language pair) would enhance clarity and completeness. We will incorporate these details into a revised version of the paper and also release the DocBlocks dataset publicly to support further research in multilingual document-level machine translation. We hope this clarifies the concerns.
>
> > It is not clearly stated which languages you work on [...] Font in figure 2 is unreadable [...] I would also expect results per language pair [...]
>
> Thank you for the feedback. We would like to clarify that all relevant dataset characteristics, including the full list of language pairs, are already presented in Figure 2. However, we acknowledge that the small font size may hinder readability, which could have contributed to the impression that this information was missing. We will update Figure 2 with larger, more readable fonts in the final version of the paper to ensure clarity. Additionally, we agree that providing results per language pair would offer valuable insight. We will include a detailed breakdown in the appendix of the final version.

---

> > ### Author Response · Authors · 2025-06-02
> >
> > > The related work section is minimal. [...]
> >
> > While we believe we have covered the most relevant literature on document-level MT, we note that this is a relatively underexplored area, both in terms of available datasets and modeling approaches. As such, the related work section is necessarily limited by the scope of existing research. That said, we will expand the section to better contextualize our contribution. In particular, we will provide a more detailed discussion of prior document-level MT corpora, highlighting their limitations, especially in terms of quality, and include a broader overview of document-level models and prompting techniques [4,5]. Having said that, we welcome any further suggestions you may have about related work you would like to see discussed.
> >
> > > I think the claim based on Figure 7, that your document training does not hurt sentence-level decoding, is exaggerated and not clearly supported by the results. Especially, IWSLT xx-en is hurt quite a lot at 10%. Also, the coloring is misleading, with different datasets in the left and right plots having the same colors. Please use different colors for different datasets.
> >
> > We respectfully disagree. Our intention was to convey that sentence-level performance is largely preserved during document-level training. This is supported by the <0.5 COMET drop across all datasets, as shown in Table 5.
> >
> > Regarding IWSLT xx→en, the observed drop at 10% occurs in the document-level performance (right plot of Figure 7), not in the sentence-level performance (left plot). This result shows that including a small amount of sentence-level data during training allows models to retain their sentence-level translation quality without severely hurting document-level translation performance. We acknowledge that using identical color schemes across the two plots may have caused confusion, and we will revise the figure to use distinct colors for each dataset.
> >
> > > I don't understand how Figure 4 is supposed to show the advantage of CAPT (which is not even mentioned in its caption), or how it relates to other results.
> >
> > Thank you for this helpful observation. We agree that the caption for Figure 4 should more clearly highlight its purpose. Context-Aware Prompt Tuning (CAPT) is a training method designed to enable models with relatively short context windows to leverage preceding translated chunks as input context during decoding. CAPT improves translation quality specifically within the contextual chunking paradigm by exposing the model to such context during fine-tuning.
> >
> > Figure 4 demonstrates the benefit of CAPT by showing improved performance of our DocMT models under contextual chunking.  We will revise the caption to explicitly mention that the figure evaluates contextual chunking performance and that CAPT is the training method responsible for the gains shown.
> >
> > Regarding its relation to other results:
> > - Figure 6 presents a comparison across translation paradigms, including Document-to-Document, Contextual Chunking, and Agent-based methods, showing that our models achieve strong translation quality while offering significantly better efficiency and scalability than agent-based methods, making them a practical and robust solution for document-level translation.
> > - Table 6 further demonstrates CAPT’s contribution, showing that incorporating CAPT alongside MRD2D on top of the curated DOCBLOCKS dataset improves performance across all translation paradigms: Document-to-Document, Chunking, and Contextual Chunking.
> >
> >
> > We appreciate the reviewer pointing this out and will ensure the connections are clearer in the final version.
> >
> > > I don’t understand Figure 6. What are all the colors supposed to illustrate? Different variants of the same model? If so, how are your variant models trained?
> >
> > Thank you for raising this point. In Figure 6, the colors represent different translation paradigms, not different models.
> > - Blue: Document-to-Document (Doc2Doc) translation, including our fine-tuned models and baselines from the WMT 2024 Shared Task on Discourse-Level Literary Translation [3].
> > - Orange: Agent-based methods — TRANSAGENTS and DELTA.
> > - Green: Contextual Chunking with our models.
> >
> > We acknowledge and will correct the labeling error: the first two agent lines should reflect the base models used -- GPT-4 Turbo for TRANSAGENTS and GPT-4o-mini for DELTA.
> >
> > We hope this clarifies.
> >
> > [1] https://aclanthology.org/2023.wmt-1.3
> >
> > [2] https://arxiv.org/pdf/2412.11732
> >
> > [3] https://arxiv.org/pdf/2405.11804
> >
> > [4] https://arxiv.org/abs/1912.08494
> >
> > [5] https://arxiv.org/pdf/1901.09115

---

> > > ### Comment · Reviewer_BHRV · 2025-06-03
> > > **Reviewer reply**
> > >
> > > Thank you very much for your response, which did clarify some of my issues.
> > >
> > > My comment based on Figure 7 was erroneous. I had misread the figure and misinterpreted your findings. I acknowledge that your interpretation is correct.
> > >
> > > Regarding the usefulness of DocBlocks, I do agree that compiling and cleaning datasets is a contribution, and I welcome the addition of additional statistics after your cleaning (which I find vital). I acknowledge that I had overlooked the comparison between the raw data and DocBlocks in Table 6. However, I donät agree with your interpretation of these scores. The differences between the first two lines, DocBlocks and the unfiltered corpus, are in most cases minor, and when DocBlocks is superior, it only slightly beats the original data. In some cases, the unfiltered data is even slightly better than DocBlocks. It is only when you add your additional methods (the three lines) that you substantially beat the unfiltered data. Presumably, the original data would also benefit from your new methods.
> > >
> > > As for the evaluation, I agree that Bleu has served well as a de facto standard. I also see the challenges with human evaluation, but I still stand by my comment that you could have performed a small human analysis on a subset for at least one language pair. While overall human evaluation can be quite subjective, as you point out, it would have been possible, for instance, to do an analysis of some discourse phenomena, such as coreference or connective translation. This would have strengthened the paper.
> > >
> > > All together, I will stand by my initial scores.

---

### Official Review · Reviewer_QBKw · 2025-05-12

**Rating:** 7
**Confidence:** 5
**Ethics Flag:** 1

**Summary:**

This paper focuses on document-level machine translation. The authors first curate a document-level translation dataset called DocBlock and then propose a multi-task instruction-tuning framework. They also explore various inference methods for the instruction-tuned model. Extensive experimental results demonstrate that the proposed approach significantly enhances the performance of base sentence-level LLMs (TowerLLM, EuroLLM, Qwen2.5) and improves their ability to leverage contextual information for translation.

**Questions To Authors:**

1. Lines 120-122: The description here is somewhat vague. Could the authors clarify what is meant by "continuing fine-tuning on document-level data"?
2. Lines 165-172: There may be potential issues with the baselines. As noted by the authors, the base models (TowerInstruct and EuroLLM) have limitations in document-level processing due to their training pipelines. It would be helpful to include results from stronger, more recent models in the camera-ready version.
3. Figure 3: What does the "D" on the right side of the x-axis represent?

**Reasons To Accept:**

1. The experiments are well-structured and effectively demonstrate the advantages of the proposed method for document-level machine translation. Additional analyses on inference methods and sentence-level translation provide a comprehensive understanding of the behavior of fine-tuned models.
2. The curated multilingual document-level dataset, DocBlock, is a valuable contribution to the community and will facilitate future research in this area.
3. The paper is well-written and easy to follow.

**Reasons To Reject:**

1. The primary concern lies in the reliability of the evaluation presented in the paper. In the main results (Table 2), some metrics appear contradictory. For instance, in the last row, Qwen2.5-7B-Instruct performs significantly worse than DocMT-Qwen2.5-7B-Instruct on BLEU, yet it surpasses it on COMET and ties on d-COMET. Incorporating human evaluations would strengthen the reliability and comprehensiveness of the results.

---

> ### Author Response · Authors · 2025-06-02
>
> We thank the reviewer for the constructive feedback and their positive assessment of our work, particularly the recognition of the value of our DocBlock dataset, the thoroughness of our experimental analysis, and the clarity of our writing.
>
> > The primary concern lies in the reliability of the evaluation presented in the paper. In the main results (Table 2), some metrics appear contradictory. For instance, in the last row, Qwen2.5-7B-Instruct performs significantly worse than DocMT-Qwen2.5-7B-Instruct on BLEU, yet it surpasses it on COMET and ties on d-COMET. [...]
>
> We appreciate the reviewer’s careful observation. The discrepancy noted is a well-documented and ongoing challenge in MT evaluation. While BLEU focuses on n-gram matching across the document, COMET captures semantic adequacy, which often leads to different preferences, easily observed in MBR decoding [1]. As mentioned in the paper (Lines 195–197), developing more robust and reliable document-level metrics remains an open problem and important direction for future work.
>
> Currently, there is no single dominant metric for document-level MT. Much like sentence-level MT a few years ago, the field is transitioning from lexical metrics like BLEU toward neural metrics such as COMET, which better correlate with human judgments. In this evolving landscape, we report a diverse set of metrics—d-BLEU, COMET, d-COMET, and LTCR—to provide a balanced and transparent evaluation. While d-BLEU remains widely used and interpretable, COMET-based metrics offer complementary insights into discourse and semantic quality. Occasional contradictions between metrics—such as the one in Table 2—are expected and reflect the inherent complexity of evaluating long-form translation [2]. This is why, instead of relying on a single metric, we opt to report a comprehensive suite of metrics at both the sentence and document levels.
>
> Ultimately, our contribution is not to claim superiority on any single metric, but to demonstrate a general and effective approach for adapting sentence-level MT models into document-level systems. The consistent gains over all base models validate the robustness and versatility of our method, even if sporadically an individual metric might counter the trend.
>
> > [...] Incorporating human evaluations would strengthen the reliability and comprehensiveness of the results.
>
> We agree that human evaluation is valuable; however, performing meaningful human evaluation for document-level MT is currently very challenging for two reasons. First, there is no standardized evaluation framework that adequately addresses the ambiguity and subjectivity involved in assessing long-form translations. For example, in IWSLT2017, where each document is a full TED talk, evaluating an entire talk's translation requires considering coherence, style, discourse, and fidelity across dozens of paragraphs, making consistent judgments highly challenging. Second, the cost of such evaluations is quite significant: it would require expert annotators to read and compare entire documents in multiple language pairs, making it impractical at scale. While we recognize its value, we believe that advancing automated document-level metrics (like d-COMET and LTCR) remains the more scalable and actionable direction for now.
>
> > The description here is somewhat vague. Could the authors clarify what is meant by ‘continuing fine-tuning on document-level data’?
>
> We appreciate the request for clarification. “Continuing fine-tuning” refers to a second-stage supervised fine-tuning process applied to models that have already been instruction-tuned—either on sentence-level MT data or other general-purpose data. In this stage, we continue the fine-tuning on document-level translation tasks using the curated DocBlocks dataset. This includes instruction-based examples in multiple formats: contextual blocks, full-document pairs, and sentence-level pairs. The goal is to adapt the models to longer contexts while preserving their sentence-level performance.
>
> We will clarify this terminology more explicitly in the revised version.

---

> > ### Author Response · Authors · 2025-06-02
> >
> > > There may be potential issues with the baselines. [...] It would be helpful to include results from stronger, more recent models in the camera-ready version.
> >
> > We agree that comparisons to stronger models are valuable for assessing overall competitiveness. While we already include strong baselines such as GPT-4o and Qwen2.5-72B-Instruct (Tables 2–3), our goal was to demonstrate that even weaker instruction-tuned sentence-level models (e.g., Tower, EuroLLM, Qwen2.5-7B) can be transformed into highly effective document-level MT systems through our approach.
> >
> > Starting from weaker baselines is a deliberate design choice—not a limitation—as it highlights the strength of our method in significantly boosting underperforming models to performance levels that rival or even exceed much larger models. This is especially valuable from a resource-efficiency perspective.
> >
> > Moreover, at the time of submission, few open-source models combined strong MT capabilities with long context windows. Tower is among the most suitable candidates for long-document translation but lacks document-level tuning—a gap our work explicitly addresses.
> >
> > Ultimately, our method provides a generalizable and practical recipe for extending sentence-level LLMs to document-level MT using high-quality data (DocBlocks) and strategies like MRD2D and CAPT. That said, we appreciate the suggestion and plan to include additional comparisons in the final version—e.g., with the recently released Qwen3.
> >
> > > What does the 'D' on the right side of the x-axis represent?
> >
> > Thank you for pointing this out. The “D” denotes “Document-to-Document” inference mode, where the entire document is translated in a single pass, contrasting with chunked inference. We will clarify this label in the figure caption in the final version.
> >
> > We hope our response addresses your main concerns.
> >
> > [1] https://aclanthology.org/2022.tacl-1.47
> >
> > [2] https://arxiv.org/abs/1912.08494

---

> > > ### Comment · Reviewer_QBKw · 2025-06-11
> > > **Re: Official Comment by Authors**
> > >
> > > Thank you for your response. I maintain my rating for this paper—it is a good piece of work.

---

### Official Review · Reviewer_cWq6 · 2025-05-13

**Rating:** 6
**Confidence:** 5
**Ethics Flag:** 1

**Summary:**

The paper presents LLM-based document translation through supervised fine-tuning. The authors explore the adaptation of sentence-level LLMs for multilingual document-level translation and introduce DOCBLOCKS, a curated dataset containing full documents and contextual segments from diverse corpora.

**Reasons To Accept:**

The creation of a new document-level data, DOCBLOCKS, represents a significant advancement for the promotion of document-level translation. The organization of data at the document level enables targeted experimentation in document-level translation.

Fine-tuning experiments highlight the effectiveness of adapting general-purpose models to specialized datasets, showcasing the potential for improved performance through contextual learning.

**Reasons To Reject:**

The distinction between results of language pairs is not exploited in the analysis of the results. Given the linguistic diversity of the evaluated language pairs, a family-wise analysis could have offered deeper insights into the models' strengths and weaknesses.

The absence of the Llama model is a notable omission. It provides support in almost 8 languages and only misses a few language pairs used in this paper.

Why was only GPT-4o and Qwen selected as baseline models for translation output? What are its advantages compared to other models? Without a clear explanation of their comparative advantages or relevance, this choice appears arbitrary. And is there a lack of discussion on the selection criterion for other models?

---

> ### Author Response · Authors · 2025-06-02
>
> We thank the reviewer for their insightful feedback and for recognizing the contributions of our work, particularly the introduction of the DocBlocks dataset and the effectiveness of our fine-tuning strategy.
>
> > The distinction between results of language pairs is not exploited in the analysis of the results. [...] a family-wise analysis could have offered deeper insights [...]
>
> Thank you for suggesting a family-wise analysis. Our aim in the current version was to highlight the broad multilingual effectiveness and generalizability of our approach across diverse language pairs. We agree, however, that a more fine-grained analysis may offer valuable insights into model performance across specific language families. In the final version, we will include an extended family-wise analysis using our existing evaluation setup with IWSLT2017 and GuoFeng. We believe this addition could help us identify the strengths and weaknesses of our models more precisely.
>
> > The absence of the Llama model is a notable omission. It provides support in almost 8 languages and only misses a few language pairs used in this paper.
>
> We appreciate the observation and understand the interest in including a LLaMA model, given its popularity. However, our model selection was primarily driven by the goal of leveraging LLMs with state-of-the-art translation capabilities, rather than starting from general-purpose backbones like LLaMA, which would require extensive adaptation for machine translation.  Our experiments include TowerInstruct-Mistral-7B [1], a model explicitly optimized for translation tasks using Mistral-7B as the backbone. Notably, it outperforms both TowerInstruct-7B and TowerInstruct-13B, LLM-based translation systems built on LLaMA.
> We will clarify these choices in the final version of the paper.
>
> > Why was only GPT-4o and Qwen selected as baseline models for translation output? [...]
>
> We selected GPT-4o and Qwen2.5-72B-Instruct as baselines for the following reasons: GPT-4o is currently the most capable publicly accessible commercial LLM for document-level translation tasks [2,3], while Qwen2.5-72B-Instruct is among the top-performing open-source multilingual LLMs. Both models support extended context windows (128K tokens for GPT-4o and 32K tokens for Qwen2.5-72B-Instruct), which are essential for document-level machine translation. This selection enables us to evaluate performance across both open-source and closed-source paradigms.
>
> > [...] And is there a lack of discussion on the selection criterion for other models?
>
> We selected Tower, EuroLLM, and Qwen2.5 (7B) as training models based on the following considerations. All three are strong sentence-level MT LLMs and provide a solid foundation for exploring extensions to document-level translation. Tower, built on Mistral, originally supported long contexts but lost its capability due to extensive continued pretraining [4] and instruction tuning in the Tower recipe, which primarily focuses on sentence-level translation tasks. EuroLLM offers broad multilingual coverage but is limited to a 4K token context window, making it a natural candidate for extension, as it was never exposed to long-context tasks. Qwen2.5 supports long-context inputs, but its 7B parameter version currently falls short in delivering high-quality document-level MT. We believe this selection offers a balanced and informative basis for studying how to generalize diverse MT systems to the document level.
>
> We hope that our answer clarifies your main concerns. We are happy to address any other question you may have.
>
> [1] https://arxiv.org/abs/2402.17733
>
> [2] https://arxiv.org/pdf/2412.11732
>
> [3] https://arxiv.org/abs/2502.12404
>
> [4] https://arxiv.org/abs/2503.19206

---

> > ### Comment · Reviewer_cWq6 · 2025-06-10
> >
> > Thank you very much for your response.
> >
> > I understand your selection of GPT-4o and Qwen-2.5-72B-Instruct as baselines, and I suggest including these details in the final version of your paper. While the Llama models may perform lower than TowerInstruct-7B, adding their scores would provide valuable reference points for the research community.
> >
> > Regarding the other models used, including a clear and thorough explanation of your selection criteria would help make the comparisons more transparent and fair. Similarly, including a study on family-wise pairs would offer deeper insights into the strengths of models.
> >
> > That said, I will keep my scores unchanged.

---

### Decision · Program_Chairs · 2025-07-08

**Decision:**

Accept

**Comment:**

The paper presents a document-level translation dataset across multiple languages.  This is constructed from several already existing publicly available document-level datasets.

Though a significant amount of work has been spent on cleaning the datasets as a curation of the DOCBLOCKS dataset presented in the paper, it is not clear that the result corresponds to improved system training in comparison to the uncurated datasets, at least in for these systems.

However, the paper also presents an approach to document-level fine-tuning which leads to good results.

The paper is well-written, the results are thorough.